# Development of Machine-Learning-Based Facial Thermal Image Analysis for Dynamic Emotion Sensing

**DOI:** 10.3390/s25175276

**Published:** 2025-08-25

**Authors:** Budu Tang, Wataru Sato, Yasutomo Kawanishi

**Affiliations:** 1Graduate School of Informatics, Kyoto University, Yoshida-Honmachi, Sakyo, Kyoto 606-8507, Japan; tang.budu.53s@st.kyoto-u.ac.jp; 2Psychological Process Research Team, Guardian Robot Project, RIKEN, 2-2-2 Hikaridai, Seika-cho, Soraku-gun, Kyoto 619-0288, Japan; 3Multimodal Data Recognition Research Team, Guardian Robot Project, RIKEN, 2-2-2 Hikaridai, Seika-cho, Soraku-gun, Kyoto 619-0288, Japan; yasutomo.kawanishi@riken.jp

**Keywords:** emotional arousal, deep learning, facial thermal imaging, machine learning, pixel-level analysis

## Abstract

Information on the relationship between facial thermal responses and emotional state is valuable for sensing emotion. Yet, previous research has typically relied on linear methods of analysis based on regions of interest (ROIs), which may overlook nonlinear pixel-wise information across the face. To address this limitation, we investigated the use of machine learning (ML) for pixel-level analysis of facial thermal images to estimate dynamic emotional arousal ratings. We collected facial thermal data from 20 participants who viewed five emotion-eliciting films and assessed their dynamic emotional self-reports. Our ML models, including random forest regression, support vector regression, ResNet-18, and ResNet-34, consistently demonstrated superior estimation performance compared to traditional simple or multiple linear regression models for the ROIs. To interpret the nonlinear relationships between facial temperature changes and arousal, saliency maps and integrated gradients were used for the ResNet-34 model. The results show nonlinear associations of arousal ratings in nose = tip, forehead, and cheek temperature changes. These findings imply that ML-based analysis of facial thermal images can estimate emotional arousal more effectively, pointing to potential applications of non-invasive emotion sensing for mental health, education, and human–computer interaction.

## 1. Introduction

Thermal imaging technology has emerged as a powerful tool for analyzing physiological and emotional states, offering significant advantages as a non-invasive, contact-free method that is particularly valued in medicine and human–computer interactions [1,2]. Unlike traditional visible-light-based imaging that can be hindered by inadequate or excessive lighting, thermal imaging performs consistently regardless of the illumination conditions, capturing infrared radiation from the body to analyze physiological changes [3]. Facial emotion recognition using visible spectrum data faces challenges due to lighting variations and captures only superficial features; thermal imaging bypasses these limitations by detecting temperature variations that are indicative of underlying physiological responses to emotional stimuli [4,5]. Temperature changes in facial skin are related to autonomic nervous system activity, particularly in the sympathetic branch that is activated during emotional arousal and causes vasoconstriction and reduced blood flow to the skin surface, leading to measurable decreases in temperature [6,7]. These temperature changes across different facial regions are valuable markers of emotional states [8,9], making thermal imaging particularly effective for applications such as mental health monitoring, education, and advanced human–computer interaction [2].

Extensive research has explored the association between facial thermal signals and subjective emotional states, using linear analyses (e.g., correlation and linear regression) based on regions of interest (ROIs) [10,11,12,13,14,15]. ROI analysis typically focuses on specific areas of the face, such as the nose tip, to determine how temperature fluctuations correspond to emotional arousal and valence. Many studies have found a consistent pattern in which the temperature of the nose tip inversely correlates with arousal levels [11,12,13]. In these studies, emotional states were assessed using the two-dimensional affective model proposed by Russell [16], in which arousal refers to the level of emotional intensity and valence to the degree of pleasantness. For example, Sato et al. found that, when participants viewed emotionally charged films, nose tip temperature dropped as subjective arousal increased, implying a physiological response to the emotional stimuli [11]. Other regions of the face that have also been studied include the forehead, cheeks, and periorbital area, although the findings have sometimes been mixed or inconclusive. Some studies have linked a reduction in forehead temperature to heightened emotional arousal [14,15], whereas another found divergent temperature changes in the cheeks and periorbital area in response to startling sounds, reflecting complex emotional processing [15]. Overall, these studies have shown that temperature shifts in different facial regions, such as the nose tip, provide valuable insights into emotional experiences.

Although previous analyses have provided valuable insights into the relationship between facial thermal signals and emotional states, they have had inherent limitations, with two major issues restricting their ability to capture detailed variation across the entire face. First, many studies have used linear analysis methods, which may not be sufficient to capture the complex, nonlinear relationships inherent in emotional processes. Linear approaches can fail to detect the subjective–physiological associations adequately, potentially failing to model the complicated neurovascular activity underlying facial temperature changes. Second, ROI-based methods rely on analyzing pre-defined facial areas, such as the nose tip or forehead, which may lead to subtle yet important thermal changes in other parts of the face being overlooked. These methods also depend heavily on manual feature extraction, which can introduce subjectivity and variability into analysis.

To address these limitations, we used pixel-level machine learning (ML) modeling, enabling examination of nonlinear associations between thermal changes across the entire face, and dynamic emotional arousal. ML techniques, particularly those capable of pixel-level analysis, offer a promising approach for assessing complex thermal features across the entire face automatically, including nonlinear relationships that traditional linear methods might miss. No study has used ML-based pixel-level dynamic emotion analysis of facial thermal images. Studies applying ML to thermal recognition of facial emotions remain relatively scarce, with most having focused on classifying discrete emotions such as happiness or fear [17,18,19]. While ML models have shown benefits in related tasks, such as thermal-to-visible image translation [20,21], their application to the continuous tracking of dynamic emotional states using pixel-level analysis of whole-face thermal images remains largely unexplored. Because there are several different ML models, we explored two conventional models—random forest [22] and support vector regression (SVR) [23]—and two deep-learning models, ResNet-18 and ResNet-34 [24].

In this study, we used ML techniques ranging from simple linear models to advanced deep-learning architectures to establish a robust framework for thermal facial emotion estimation. We recorded thermal data of participants’ faces and obtained dynamic valence and arousal ratings while they observed five emotional film clips. To assess the feasibility of predicting emotional arousal from thermal facial data, as a baseline, we began with linear regression analysis of the nose tip ROI. To perform pixel-level analysis across the entire face and address the limitations of ROI methods, we explored deep-learning models capable of capturing detailed thermal patterns (Figure 1) after image preprocessing (Figure 2). We applied leave-one-person-out cross validation (LOPOCV) to evaluate predictive performance rigorously and address concerns related to overfitting. To overcome the black-box nature of the ML models and interpret our results, we first used saliency maps to identify which facial regions had the most significant impact on the model predictions by highlighting the areas where temperature changes contributed most to emotional state estimation. Subsequently, we applied the integrated gradients method to these significant regions to analyze patterns of temperature changes in relation to emotional states. Based on ample evidence showing an association between facial thermal changes and subjective emotional arousal [11,12,13,14,15], together with the rationale that facial thermal changes reflect sympathetic nervous system activity, which is theoretically coupled with arousal [6,7], we analyzed the arousal ratings. This comprehensive approach helped identify influential facial regions and enhanced our understanding of how ML methods can be leveraged for pixel-level thermal emotion analysis.

## 2. Materials and Methods

### 2.1. Participants

This study recruited 20 healthy Japanese adults (10 women, 10 men; mean ± standard deviation [*SD*] age, 22.0 ± 2.6 years). Although an 11th woman participated, her data were not analyzed due to substantial motion artifacts. The sample size was determined based on previous research examining the relationship between facial thermal responses and emotional states [11]. All participants had normal or corrected-to-normal vision, without the use of glasses, and were native speakers of Japanese. The inclusion criteria included willingness to participate in subjective and physiological measurements, no neurological or psychiatric disorders, and no prior experience with the emotional film clips used in the study. All participants consented after being fully informed about the experimental procedures. The study was approved by the RIKEN Ethics Committee and was conducted in accordance with institutional ethical standards and the Declaration of Helsinki.

### 2.2. Apparatus

The experimental setup consisted of a Windows-based HP Z200 SFF computer (Hewlett-Packard Japan, Tokyo, Japan) running Presentation software ver. 14.9 (Neurobehavioral Systems, Berkeley, CA, USA) to control stimulus delivery. Visual stimuli were presented on a 19-inch monitor (model HM903D-A; Iiyama, Tokyo, Japan) with 1024 × 768-pixel resolution. Participants used a wired optical mouse (model MS116; Dell, Round Rock, TX, USA) connected to an additional Windows laptop (model CF-SV8; Panasonic, Tokyo, Japan) to collect dynamic ratings. Thermal imaging was conducted using an A655sc infrared thermal camera (FLIR Systems, Wilsonville, OR, USA) and the Research IR Max software (ver. 4.40). Positioned adjacent to the monitor, the camera captured full-face thermal images at a spatial resolution of 640 × 480 pixels with a frame rate of 50 Hz. Although additional thermal data of the profiles and full-face RGB data were acquired as part of the experiment, these data are not reported in this paper.

### 2.3. Stimuli

A set of five film clips was selected to elicit a range of emotional responses: “Cry Freedom” (very negative, anger), “The Champ” (moderately negative, sadness), “Abstract Shapes” (neutral), “Wild Birds of Japan” (moderately positive, contentment), and “M-1 Grand Prix The Best 2007–2009” (very positive, amusement). Several previous studies used and validated these films as a means of eliciting emotion [11,25]. Specifically, anger, sadness, neutral, contentment, and amusement films reportedly showed linear and quadratic relationships with subjective valence and arousal ratings, respectively [25]. The mean ± *SD* duration of these films was 175.8 ± 22.2 s, with individual durations of 157 s for anger, 172 s for sadness, 206 s for neutral, 148 s for contentment, and 196 s for amusement. Two additional films, depicting scenes from “The Silence of the Lambs” and “Colour Bars” from Gross and Levenson, were used for practice trials. The stimuli were presented at a 640 × 480-pixel resolution, corresponding to visual angles of approximately 25.5° horizontally and 11.0° vertically.

### 2.4. Procedure

The experiments were conducted in a soundproof, electrically shielded chamber. The ambient temperature was maintained at between 23.5 °C and 24.5 °C and was monitored using a TR-76Ui data logger (T&D Corp., Matsumoto, Japan). Participants were informed that the purpose of the study was to obtain subjective emotional ratings and record physiological responses while they viewed a series of films. The subjects were provided with approximately 10 min to acclimate to the room conditions before the experiment commenced.

Participants were seated comfortably on a chair positioned approximately 0.77 m from the monitor. The thermal imaging camera was placed adjacent to the monitor to capture full-face thermal images continuously throughout the experiment. Following two practice films to familiarize participants with the procedure, the five test films were presented in a pseudorandom order.

Each trial began with a 1-second fixation point displayed at the center of the screen, followed by a 10-second white screen that served as a pre-stimulus baseline. The film clip was then presented. The onset of thermal image acquisition was synchronized with the film presentation, and data were acquired during the pre- and post-stimulus periods. After the film concluded, another white screen was displayed for 10 s as a post-trial baseline. Participants were then asked to rate the overall subjective experience of emotional valence and arousal using an affect grid [16], which provided a nine-point scale for each dimension. The ratings were entered by pressing the number keys 1–9 on the keyboard, corresponding to the participant’s perceived level of valence and arousal. Each participant was instructed to fixate on the central point, watch each film attentively, and subsequently rate the subjective experience by pressing the appropriate number keys. After the rating was provided, the screen turned black during the inter-trial interval, which varied randomly between 24 and 30 s. Thermal imaging data were recorded continuously throughout all trials, along with digital markers indicating the onset of each film.

Upon completion of all trials, participants engaged in a dynamic rating session. All film stimuli were presented again on the monitor. Participants were instructed to recall their subjective emotional experiences during the initial viewing and continuously rate their experiences in terms of valence and arousal by moving the mouse. The mouse coordinates were sampled at a rate of 10 Hz. The onset of rating data acquisition was synchronized with film presentation. This cued recall procedure was used to obtain continuous ratings of valence and arousal, which were challenging to collect simultaneously during the initial viewing. Previous studies have indicated that cued recall continuous ratings are strongly correlated with real-time continuous ratings for emotional films [26,27].

### 2.5. Workflow

The workflow used for analyzing facial thermal images to estimate emotional responses with the ResNet-34 model is illustrated in Figure 1. The process involved acquiring thermal images, followed by facial registration using UV mapping for consistent alignment. Preprocessing included random transformations and normalization, and the data were analyzed using LOPOCV, which evaluates both linear regression and deep learning models. Saliency maps and integrated gradients were used to interpret the model results, highlighting important facial regions and their contributions to emotional states.

### 2.6. Preprocessing

The first thermal frame was imported into the Blender 3.6 software; its mesh was then manually rotated and translated until it matched a standardized three-dimensional (3D) facial template. Blender then generated a UV map that defined a fixed correspondence between image pixels and template vertices (Figure 2). This reference UV map was exported and applied to every subsequent frame. To propagate the alignment automatically, we used our landmark-tracking script implemented in MATLAB R2020a (Mathworks, Natick, MA, USA) as follows: facial landmarks were detected in each frame, and their coordinates were used to estimate a rigid (rotation + translation) transform that brought the frame into the reference coordinate system before re-sampling onto the template surface. This two-step procedure combined manual adjustment with automatic landmark-based alignment, allowing the method to compensate for small head movements and changes in facial orientation, thereby reducing motion-related variability across the entire sequence.

After registration, the facial region was cropped to remove extraneous areas that did not contribute to the analysis, such as the background or neck. Cropping focuses the analysis on the facial regions that are most relevant for emotional detection, reducing computational requirements and improving model accuracy by eliminating irrelevant information.

Normalization was then performed to ensure consistent pixel intensity values across the dataset. This step reduced the influence of varying thermal conditions that might arise from differences in ambient temperature or individual physiological variation. By scaling the intensity values to a consistent range, the models can focus on meaningful differences in temperature across facial regions, which are indicative of emotional responses.

A smoothing filter was also applied to reduce noise in the thermal images. Given that thermal cameras can sometimes produce noisy images, particularly in low-resolution settings, smoothing helps to enhance the signal-to-noise ratio. A Gaussian filter was commonly used for this purpose, providing a balance between reducing noise and preserving important features in facial regions.

These preprocessing steps collectively ensure that the thermal images are of high quality and consistently aligned, providing a solid foundation for subsequent analysis and modeling.

### 2.7. Model Training and Validation

The analysis of facial thermal images to predict emotional responses involves training and validating several models, ranging from simple linear models to those with more complex deep learning architectures.

To establish a baseline for predicting emotional arousal, linear regression was used as a straightforward yet effective starting point. Two types of linear models were implemented: one using only the nose tip ROI and another using multiple ROIs, including the nose tip, forehead, and bilateral cheeks.

Following the baseline, two conventional ML models—random forest regression and SVR—were evaluated. For the random forest model, we used the RandomForestRegressor from scikit-learn with 300 decision trees, a maximum depth of 15, and a minimum leaf size of 2. The SVR model was implemented with a radial basis function kernel, and its hyperparameters were optimized using grid search. For these analyses, thermal images were resized to 64 × 64 pixels, converted to grayscale, and flattened into 4096-dimensional vectors before being processed. To reduce noise and improve efficiency, a principal component analysis was conducted to reduce the dimensionality to 200 before training.

We then explored various deep learning architectures, including lightweight models such as MobileNet, but their performance differences were marginal. Based on its consistently superior accuracy, we ultimately selected a regression model with a ResNet-34 backbone for the main analyses. A smaller ResNet-18 model was also included for comparison, serving as a lightweight convolutional neural network alternative. Both models were implemented using the torchvision library with ImageNet-pretrained weights. Their final fully connected layers were replaced with a single output node to support continuous arousal prediction. The input images were converted to RGB, resized to 224 × 224 pixels, and normalized using the standard ImageNet mean and standard deviation. The models were trained using the Adam optimizer (learning rate = 1 × 10^−4^; weight decay = 1 × 10^−4^), a batch size of 32, and the mean squared error (MSE) loss for 10 epochs. The process began with image acquisition, followed by preprocessing steps including facial registration to ensure consistent alignment across frames. Within the ResNet backbone, the thermal images passed through multiple residual blocks to extract discriminative features associated with emotional arousal. Each residual block typically comprises a convolutional layer, a batch normalization (BN) layer, and a rectified linear unit (ReLU) activation function. The convolutional layer uses learnable filters to scan local receptive fields, capturing the spatial patterns that are crucial for downstream predictions. Next, the BN layer standardizes the per-channel outputs within each mini-batch to attain near-zero mean and unit variance, thereby mitigating internal covariate shift and stabilizing gradient flow. The ReLU activation function then zeroes out negative values while preserving positive ones, which introduces nonlinearity and helps to prevent vanishing gradients. Meanwhile, each residual block integrates a skip connection that sums its input directly with its output, improving gradient propagation and enabling deeper architectures. After the final residual block, a pooling layer reduces the spatial dimensions of the resulting feature maps, which are then flattened into a one-dimensional vector. To reduce overfitting further, a dropout layer randomly deactivates a fraction of neurons during each training iteration, thereby promoting more robust feature learning. Finally, an output layer produces a continuous value that reflects the predicted emotional arousal from the facial thermal image.

To evaluate the models, LOPOCV was applied. In this approach, each participant’s data were left out once as a test set, while the remaining participants’ data were used for training. This enables a robust evaluation of model generalizability, ensuring that the trained model is not overly reliant on the data of any single participant. Both the linear regression and lightweight deep-learning models were evaluated, with metrics such as the correlation coefficients and MSE used to compare their predictive accuracy.

### 2.8. Statistical Analysis

The mean dynamic ratings during film presentation (five ratings for each participant) were subjected to repeated-measures trend analyses (two-tailed). The linear and quadratic natures of the valence and arousal ratings were assessed across five films (anger, sadness, neutral, contentment, and amusement) to confirm the previous finding [25].

To validate the effectiveness of the models, we performed statistical analyses to compare the predictive accuracy of the linear regression and deep learning models. Using the LOPOCV approach, a separate model was trained for each participant, and the correlation between the predicted and actual arousal ratings was computed for each individual. To evaluate the prediction performance, the correlation coefficient produced by each model was analyzed using a one-sample *t*-test contrasting with zero (two-tailed). To determine whether there were significant differences in the correlations obtained by the models, one-way repeated-measure analyses of variance (ANOVAs) with model as a factor, followed by multiple comparisons using the Bonferroni method (two-tailed), were performed. We also evaluated the MSE using these tests.

The α-level for all analyses was set to 0.05.

### 2.9. Model Interpretation

To gain insights into the key facial regions contributing to emotional state predictions and the nonlinear relationships between facial temperature changes and arousal, we used a two-step interpretability approach for the ML models with saliency maps [28], followed by integrated gradients [29]. We analyzed the ResNet-34 model, which showed the highest estimation performance in the statistical tests.

First, saliency maps were generated to identify the areas of the face that had the greatest impact on the model predictions. By highlighting the pixels contributing most to the output, these maps provided an overview of important facial regions, such as the nose tip, forehead, and cheeks, which were identified as critical indicators of emotional arousal.

After identifying these key regions using saliency maps, we applied integrated gradients to conduct a more detailed analysis of these specific areas. The integrated gradients allowed a quantitative assessment of the contribution of each pixel by computing the path integral of gradients as the input transitioned from baseline to its actual value. This allowed us to measure the magnitude of the effect that temperature changes in the identified facial regions had on the predicted arousal values, offering a more precise understanding of the nonlinear relationships captured by the deep learning models. To evaluate nonlinear relationships between temperature and arousal ratings, polynomial regression analysis was performed using first-degree (linear), second-degree (quadratic), third-degree (cubic), and fourth-degree (quartic) models. The optimal model was selected based on the adjusted R^2^, root mean squared error (RMSE), Akaike’s information criterion (AIC), and Bayesian information criterion (BIC).

## 3. Results

### 3.1. Subjective Ratings

Figure 3 shows the group mean time courses of the dynamic valence and arousal ratings. The figures indicate that the emotional film clips elicited dynamic changes in subjective valence and arousal. For example, the valence curve for the angry film showed a slight increase followed by a sharp decline, reflecting the film’s content: a pleasant group gathering scene followed by an anger-inducing massacre. Planned contrasts confirmed that the mean dynamic ratings acquired during film presentation reflected the expected linear and quadratic patterns of the valence and arousal ratings across films, respectively (valence: *t*[76] = 13.45, *p* < 0.001; arousal: *t*[76] = 9.35, *p* < 0.001).

### 3.2. Estimation Performance

Figure 4 presents a representative arousal trajectory for a held-out participant in the LOPOCV test, comparing the ResNet-34 model’s predictions with the participant’s self-reported ratings and illustrating their temporal correspondence.

Figure 5 summarizes the mean ± standard error of the MSE and the Pearson correlation coefficients between the predicted and actual arousal ratings obtained using six arousal prediction models: two baseline ROI-based linear regression models (simple and multiple regression models), two conventional ML models (random forest regression and SVR), and two deep-learning models (ResNet-18 and ResNet-34). The evaluation was based on LOPOCV, in which the data from one participant served as the test set and the rest were used for training.

One-sample *t*-tests verified that the correlation coefficients of all models were significantly greater than zero (t[19] > 18.9, *p* < 0.001, *d* > 1.0), confirming that each model captured meaningful arousal information.

A one-way repeated-measures ANOVA with model as a factor on the correlation coefficients revealed a significant main effect of model (*F*[5, 95] = 61.34, *p* < 0.001, η^2^_p_ = 0.76). Bonferroni-corrected multiple comparisons showed that the four ML models were significantly more correlated with the ground-truth arousal ratings than the two linear baseline models (*p* < 0.001). No significant difference was observed among the ML models (*p* > 0.05).

A parallel ANOVA on the MSE likewise yielded a significant main effect of model (*F*[5, 95] = 21.01, *p* < 0.001, η^2^_p_ = 0.53). Post hoc tests indicated that both deep learning models achieved lower MSEs than the other four models, and both conventional ML models achieved lower MSEs than the two linear regression models (*p* < 0.001).

### 3.3. Model Interpretation

To interpret the nonlinear relationships between facial temperature changes and arousal, we applied model interpretation tools. We analyzed the ResNet-34 model, which had the highest estimation performance in the above analyses.

We first used saliency maps to visualize the facial regions that had the greatest influence on the ML model predictions. Figure 6 shows the saliency map alongside a corresponding original thermal image of a representative participant. The saliency map highlights the key facial regions that had the most significant impact on predicting emotional arousal, such as the nose tip, forehead, and both cheeks.

To interpret the model predictions further, integrated gradients were used to analyze the contributions of representative facial regions, including the nose tip, forehead, and both cheeks. Integrated gradients are an attribution method used to understand the influence of each input feature on model predictions by calculating the path integral of the gradients from a baseline input to the actual input. This enables a more nuanced understanding of how individual pixel changes affect the output of the model, thus providing insights into which areas of the input contribute most significantly to the decision of the model. To quantify the visually observed patterns and rule out the possibility that apparent nonlinearities were merely a consequence of an over-parameterized model, we fitted first- to fourth-degree polynomials to the temperature–attribution pairs of each ROI and compared them with adjusted R^2^, RMSE, AIC, and BIC values (Table 1). For the nose tip, the adjusted R^2^ increased from 0.64 for the linear fit to 0.71 for the cubic fit, while the quartic term produced only a negligible gain; because AIC and BIC bottomed out at the cubic–quartic transition and the cubic model is more parsimonious, the cubic degree was selected as optimal for this ROI. The forehead showed a modest linear fit but improved sharply only when a quartic term was added, raising the adjusted R^2^ to 0.11 and lowering AIC by about 270 points relative to the cubic fit; therefore, a quartic model was selected as optimal for the forehead. Both cheeks displayed much weaker effects. The left cheek rose monotonically to an adjusted R^2^ of 0.07 at the quartic degree, with the AIC and BIC values continuing to decline; despite the small effect size, the quartic fit was selected for completeness. The right cheek reached an adjusted R^2^ of 0.10 under the cubic model and showed virtually no further improvement at the quartic degree; in this case, the cubic fit offered the best balance between fit and simplicity, and was chosen as the best model. Figure 7 illustrates these final choices by overlaying the selected polynomial curve on the frame-level scatter for each ROI.

## 4. Discussion

Our model comparisons using correlation coefficients and MSE (Figure 4) demonstrated that the ML model significantly outperformed the ROI-based linear regression model in predicting emotional arousal from facial thermal data. Although the linear regression model for the nose tip ROI showed a significant correlation between the predicted and actual arousal values, consistent with previous findings [11,12,13], the ML model had superior predictive performance for both measures. These results imply that the relationship between facial thermal signals and emotional arousal is not purely linear, instead involving complex, nonlinear dynamics that are better captured by ML models. Regarding the prediction performance of the different ML models, correlation coefficients between the predicted and actual arousal ratings were comparable between the conventional (i.e., random forest and SVR) and deep-learning (i.e., ResNet-18 and ResNet-34) models, although the MSE values of the latter models were better. These results are consistent with the findings of some previous studies that reported a comparable estimation performance between conventional and deep-learning models [30,31], although other studies reported better estimation performance by deep-learning models than by conventional models [32]. We postulate that one factor contributing to this inconsistency may be the limited sample size and relatively uncomplicated model structure employed in this study [30]. Taken together, our results demonstrate that an ML model can predict emotional arousal from thermal facial imaging accurately, underscoring its potential for emotion detection tasks.

To interpret our ML model, we used saliency maps [28] to visualize the facial regions that influenced the model predictions the most. Saliency maps help identify which areas of the input data have the greatest impact on the model output, thereby providing insights into the underlying decision-making process of an ML model. Our results show that the forehead, nose tip, and both cheeks had the highest brightness on the saliency maps, indicating that these regions contributed most significantly to the prediction of emotional arousal.

To explore the importance of these high-contribution regions further, we used integrated gradients [29], an interpretability technique that quantifies the contribution of each input feature to the model output. The average attribution value curve for the nose tip region had negative and approximately linear associations with the subjective arousal ratings, which aligns well with previous linear analyses’ results [8,11]. The temperature decrease at the nose tip and in other regions during heightened emotional arousal can be attributed to increased sympathetic nervous system activity, which induces vasoconstriction and reduces blood flow [7,8]. The absence of underlying muscles in the nose tip, unlike other facial regions, may minimize nonlinear thermal interference from muscle contractions [7,33]. The average attribution curve for the forehead showed distinct peaks and valleys across different temperature ranges, indicating significant fluctuations in its contribution to the model. Our analysis revealed a complex, nonlinear relationship between forehead temperature and emotional arousal, which has not been captured by previous studies, which primarily reported a straightforward negative correlation [8,34]. This implies that the forehead exhibits dynamic changes beyond a simple linear trend, further emphasizing its role in reflecting emotional states. Similarly, both cheeks exhibited nonlinear relationships between temperature and integrated gradients attribution. Although the left- and right-cheek trajectories shared comparable shapes, the right cheek showed somewhat larger attribution magnitudes (Figure 7). This pattern implies a broadly bilateral consistency in autonomic thermal responses during emotional arousal, with minor side-specific amplitude differences.

Our findings demonstrate the potential real-world applications of ML-based facial thermal analysis, particularly mental health monitoring and intelligent interventions. By providing a non-invasive, automated, continuous approach to assess emotional arousal, our research enables more effective and precise tracking of emotional states. Traditional methods for assessing emotion-related mental health, such as questionnaires and structural interviews, often present challenges due to their subjective nature, difficulty in acquiring quantitative measures, and continuous recording that may fail to capture the dynamic nonlinear relationships inherent in emotional and physiological data. By contrast, ML models applied to facial thermal data can continuously track emotional changes by analyzing subtle, pixel-level variation across the entire face, providing more precise and objective insights into an individual’s mental state. This high granularity in emotional detection is crucial for identifying early signs of mental health conditions, such as anxiety or depression, which may manifest via nuanced nonlinear changes in emotional arousal. Unlike conventional linear ROI-based methods that focus on specific facial areas, like the nose or forehead, ML models can learn dynamically from an entire thermal imaging dataset, allowing them to capture intricate interactions between different facial regions and their thermal signatures. This capability is particularly valuable for tailoring personalized interventions, as the system can detect and respond to minor emotional shifts, potentially alerting users or healthcare professionals before emotional dysregulation becomes pronounced.

However, several limitations of this study must be acknowledged. First, the sample size of the present study was modest (*n* = 20) and culturally homogeneous, comprising young Japanese adults recruited from a single site. This limitation may restrict the generalizability of the findings. Future work should employ stratified, multisite recruitment to assemble larger and demographically diverse cohorts across age, gender, and ethnicity, thereby enabling more precise estimates of interindividual variability and stronger external validity. Second, only five film clips were used as stimuli; so, the generalizability of the present findings remains unproven. Future research using additional film stimuli is warranted to develop a robust understanding of the association between facial thermal changes and subjective arousal. Third, our analysis relied exclusively on facial temperature and lacked multimodal validation. Future studies should collect additional physiological signals (e.g., electrodermal activity, electrocardiography, and electroencephalography) to provide independent benchmarks of autonomic nervous system activity and arousal beyond self-reports. Fourth, although ambient temperature was held at 23.5–24.5 °C, residual variation in room climate, baseline skin temperature, respiration patterns, and subtle facial muscle activity were not explicitly modelled and could have introduced noise. Incorporating these covariates or adopting adaptive compensation schemes in future protocols would further enhance predictive accuracy. Finally, while ResNet-34 offered the best performance among the tested models, exploring alternative architectures—ranging from detection-style networks like Transformer [35] to lightweight attention-based frameworks—and re-engineering them with regression heads tailored to continuous arousal estimation should help determine whether even greater accuracy and interpretability can be achieved.

## 5. Conclusions

Our study demonstrated that an ML approach significantly outperformed linear regression in predicting emotional arousal from facial thermal data, highlighting its ability to model the complex, nonlinear relationships inherent in emotional responses. By leveraging pixel-level analysis across the entire face, our ML model captured subtle thermal variation that correlated with emotional arousal, thereby achieving greater predictive accuracy as evidenced by higher correlation coefficients and lower MSE values. The use of saliency maps, along with integrated gradients, provided insights into the key facial regions in arousal prediction, emphasizing the value of ML for enhancing the interpretability and precision of emotion detection systems. Despite some limitations, such as a relatively small dataset and the focus on temperature as the sole physiological measure, our findings establish an important foundation for future research. ML shows great promise for real-world applications in emotion detection, offering a more reliable, non-invasive means of assessing emotional states; this has potential implications for mental health monitoring and adaptive interventions.

## Figures and Tables

**Figure 1 sensors-25-05276-f001:**
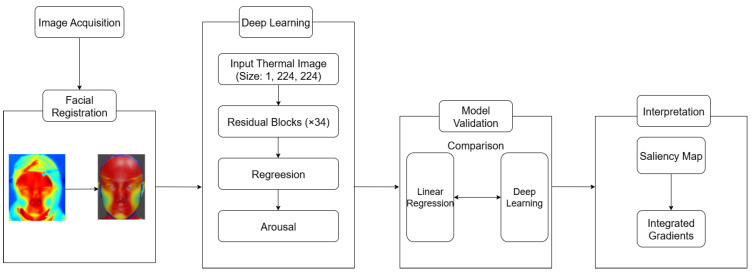
Workflow used for thermal-image-based emotion analysis with the ResNet-34 model.

**Figure 2 sensors-25-05276-f002:**
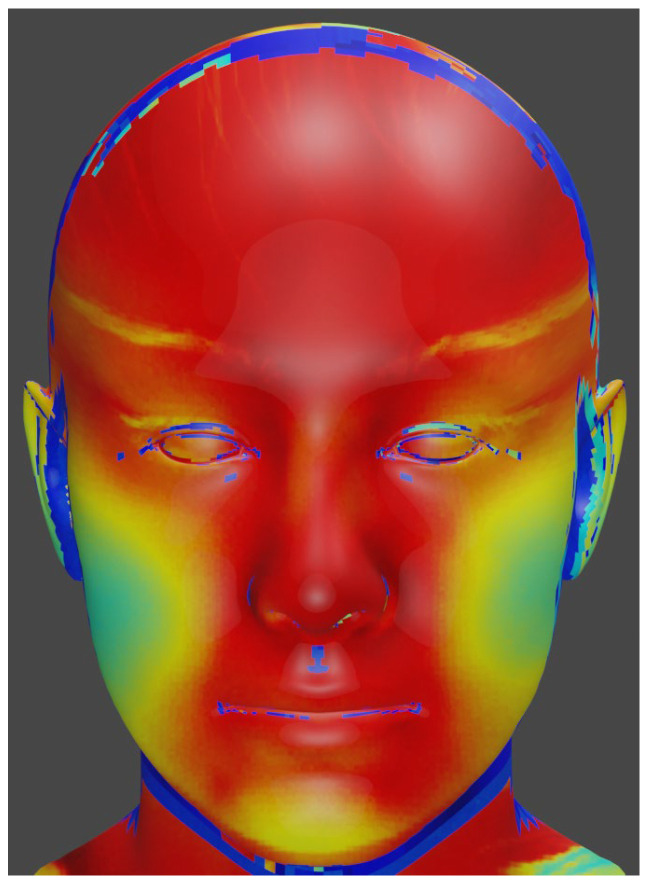
The image maps the original thermal data onto a standardized three-dimensional facial model using ultraviolet mapping.

**Figure 3 sensors-25-05276-f003:**
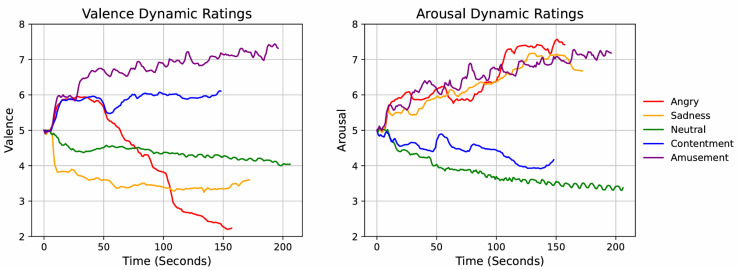
Group mean ratings of the second-by-second dynamic valence (**left**) and arousal (**right**) elicited by the emotional film clips. The plots illustrate the temporal fluctuations in subjective emotional responses, showing how valence and arousal ratings vary across film clips.

**Figure 4 sensors-25-05276-f004:**
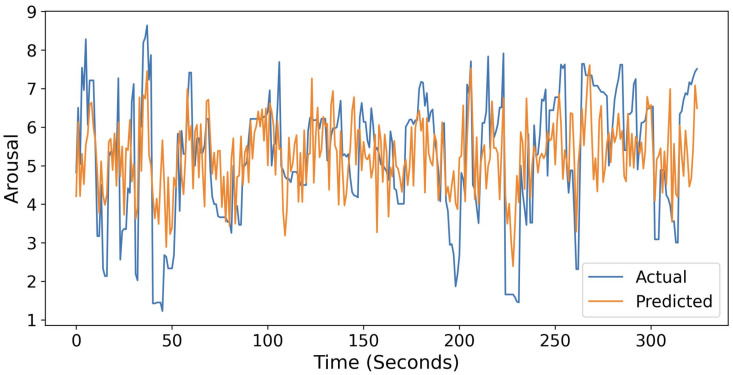
A representative example of actual and predicted arousal ratings (*r* = 0.57) obtained from the ResNet-34 model.

**Figure 5 sensors-25-05276-f005:**
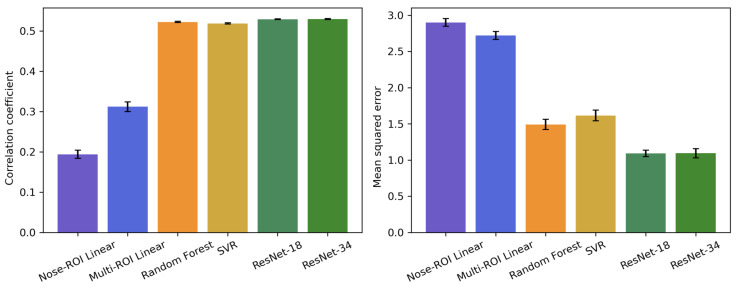
Mean ± standard error of Pearson correlation coefficients (**left**) and mean squared error (**right**) for six arousal prediction models: nose region of interest (ROI) linear regression, multi-ROI linear regression, random forest regression, support vector regression (SVR), ResNet-18, and ResNet-34.

**Figure 6 sensors-25-05276-f006:**
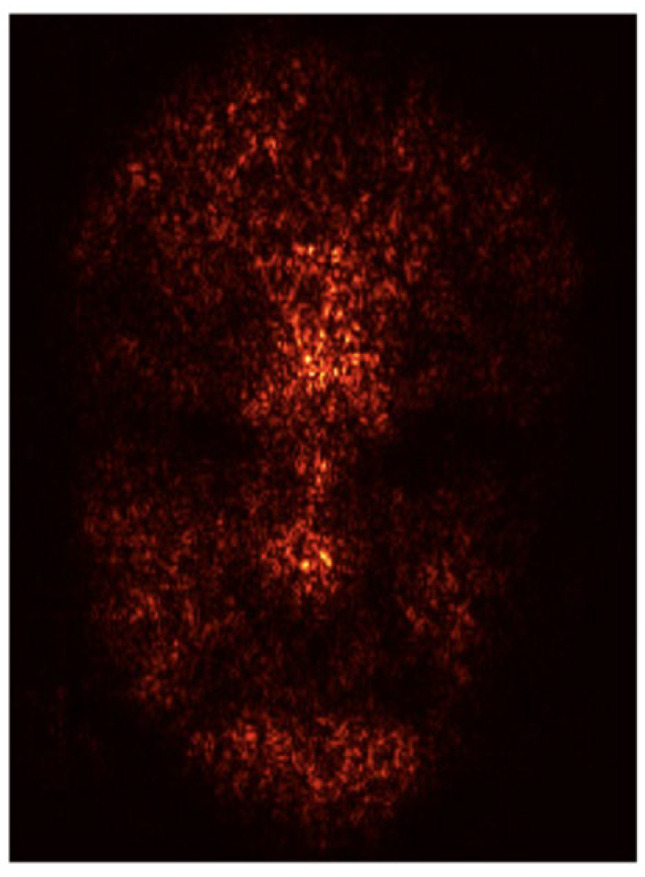
The saliency map. The brighter areas made greater contributions to the model predictions. The map is depicted using the standardized three-dimensional facial model coordinate system in Figure 2.

**Figure 7 sensors-25-05276-f007:**
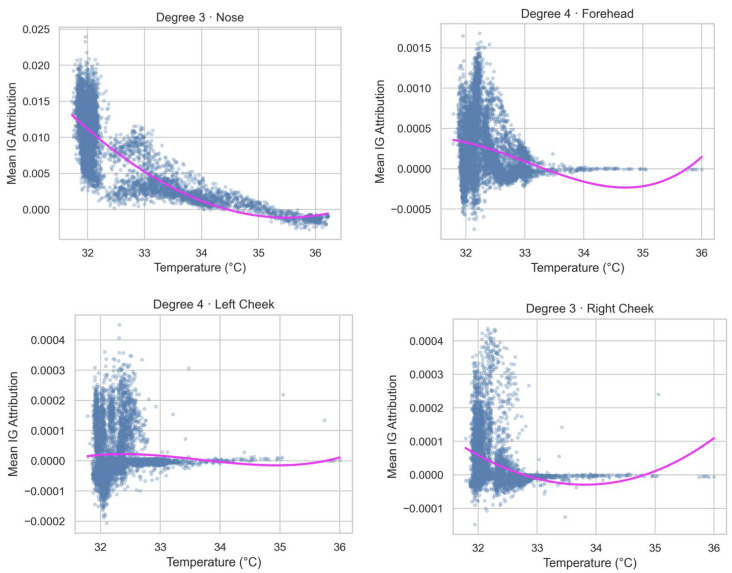
The scatterplots and polynomial regression lines of integrated gradients (IGs) for the representative facial regions, including the nose tip, forehead, and left and right cheeks, showing the nonlinear relationships between the temperature and arousal ratings in these regions.

**Table 1 sensors-25-05276-t001:** Fit indices (adjusted R^2^, root mean squared error [RMSE], Akaike’s information criterion [AIC], and Bayesian information criterion [BIC]) values for polynomial models (degrees 1–4) applied to each facial region of interest (ROI).

ROI	Degree	Adjusted R^2^	RMSE	AIC	BIC
Nose	1	0.639401	0.003378	−74,632.8	−74,619.2
	2	0.699779	0.003082	−75,833.5	−75,813.2
	**3**	**0.706937**	**0.003045**	**−75,990.8**	**−75,963.7**
	4	0.707854	0.00304	−76,010.4	−75,976.4
Forehead	1	0.058802	0.000361	−103,975	−103,961
	2	0.059797	0.00036	−103,981	−103,960
	3	0.073815	0.000358	−104,078	−104,051
	**4**	**0.111072**	**0.00035**	**−104,347**	**−104,313**
Left cheek	1	0.005772	8.08 × 10^−5^	−123,590	−123,577
	2	0.009857	8.08 × 10^−5^	−123,592	−123,571
	3	0.058239	8.06 × 10^−5^	−123,623	−123,596
	**4**	**0.06942**	**8.06 × 10^−5^**	**−123,629**	**−123,595**
Right cheek	1	0.074671	7.73 × 10^−5^	−124,177	−124,164
	2	0.097298	7.63 × 10^−5^	−124,339	−124,318
	**3**	**0.102118**	**7.61 × 10^−5^**	**−124,373**	**−124,346**
	4	0.102143	7.61 × 10^−5^	−124,372	−124,338

The optimal models are in bold.

## Data Availability

Data statistically analyzed during this study are included in this published article and its Appendix A Files. The facial thermal images are not publicly available due to privacy or ethical restrictions.

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
