# Peer review of "Development of Machine-Learning-Based Facial Thermal Image Analysis for Dynamic Emotion Sensing"

_sensors, 2025, doi:10.3390/s25175276_

Round 1
Reviewer 1 Report
Comments and Suggestions for Authors
comments and suggestions are listed in the attachment.

Author Response
Dear Sir,
Thank you for your email dated June 26, 2025, in which you kindly advised us to revise our manuscript (sensors-3716574). We have carefully revised the manuscript based on your suggestions. The major changes made to the manuscript are indicated in red font in the tracked version. Additionally, a professional English proofreading service made some minor edits (http://www.textcheck.com/certificate/lTTyPc); these changes are not highlighted unless the content itself was modified.
Point 1: The sole comparison between the ResNet-34 model and a simple linear-regression model using only the nose-tip Region of Interest (ROI) is inadequate. The established negative correlation between nose-tip temperature and arousal makes this a weak baseline. This design risks overstating the advantage of the ML approach.
Point 2: Essential controls are missing: The study fails to compare the deep-learning model against more robust baselines, such as:
- Linear models incorporating multiple relevant ROIs (e.g., nose tip + forehead + cheeks).
- Traditional non-deep-learning ML methods applied to pixel-level or feature-extracted data (e.g., SVM, Random Forest, PCA combined with regression).
- Physiological validation: Crucially, the model predicts subjective arousal ratings but lacks validation against objective physiological measures of arousal. This absence makes it impossible to discern whether the model learns true physiological correlates of emotion or spurious noise patterns unrelated to underlying arousal. Correlating thermal changes with established physiological markers is essential for validating the approach’s biological plausibility.
Response: As suggested, we have added three additional machine learning models, including random forest regression, support vector regression, and ResNet-18, along with a region-of-interest-based multiple linear regression model. The results are reported in the Results section (p. 9). Our new findings demonstrate that ML-based analysis of facial thermal images can more effectively estimate emotional arousal.
Regarding part (c), concurrent physiological recordings (e.g., electrodermal activity, heart rate variability) were not collected in the present experiment; therefore, a direct benchmark against such signals is unfortunately not possible. We have acknowledged this limitation in the revised manuscript and have suggested simultaneous physiological measurements as an important direction for future research (Discussion, p. 14).
Point 3: Limited Sample Size and Diversity: A sample of N = 20 homogeneous participants (Japanese university students, mean age 22.0 ± 2.6 years) is too small and lacks demographic diversity (age, ethnicity). This severely restricts the generalizability of the findings. Furthermore, while gender was balanced (10 F / 10 M), no analysis was presented on potential gender differences in thermal responses or model performance.
Response: We acknowledge that the present cohort of 20 Japanese university students is relatively small and demographically homogeneous. While this sample was sufficient for an initial proof-of-concept, it limits the generalizability of our findings and precludes a systematic assessment of age, ethnicity, or gender effects. In the revised manuscript (Discussion, p. 13), we now explicitly highlight this limitation and note that future studies should (i) recruit a larger and more diverse participant pool spanning multiple age groups and ethnic backgrounds, and (ii) be designed with adequate statistical power to examine potential sex-related differences in thermal-emotion responses and model performance.
Point 4: Restricted Stimulus Set: Using only 5 film clips to elicit a range of emotional states provides limited coverage of the emotional spectrum, particularly high-arousal states like fear. The reliance on previous validation without reporting specific elicitation success rates within this study cohort is a weakness.
Response: As suggested, we have discussed the limitation of using only five film clips in the Discussion section (p. 13).
Additionally, we have reported the results of planned contrasts, which confirm the expected patterns of subjective ratings in the Results section (p. 8).
Point 5: Inadequate Control for Motion Artifacts: Thermal imaging is highly sensitive to head movement. While UV mapping for registration is mentioned, the manuscript lacks details on how motion artifacts were quantified, controlled, or corrected beyond registration (e.g., using optical-flow tracking or specific motion-correction algorithms). Residual motion noise could significantly confound the pixel-level temperature signals attributed to emotion.
Response: We have clarified our motion-control procedure in the revised manuscript (p. 5). Specifically, we combined manual UV alignment in Blender for the first frame with automatic landmark-based alignment in MATLAB for all subsequent frames. This two-step process compensates for head movements and ensures consistent spatial alignment across the entire sequence.
Point 6: Superficial Attribution Analysis: While saliency maps and Integrated Gradients are used, the interpretation remains descriptive (highlighting nose, forehead, cheeks). There is no quantitative assessment of the relative contribution of these regions to the model’s prediction. The claim of nonlinear relationships (especially for the forehead) based on Integrated-Gradient plots lacks statistical validation against simpler linear or alternative nonlinear models within those specific ROIs and could reflect overfitting rather than a genuine physiological phenomenon.
Response: To provide a rigorous and quantitative complement to the saliency and Integrated Gradients visualizations, we have now included a complete polynomial model comparison for each facial ROI. Specifically, we fitted linear, quadratic, cubic, and quartic regressions to the temperature-attribution pairs for the nose tip, forehead, and both cheeks, and reported adjusted R2, RMSE, AIC, and BIC for each fit (Table 1, p. 11). The key findings are summarized in Figure 6 (p. 10). For the nose, the adjusted R2 increased from 0.64 (linear) to 0.71 under the cubic model, with only a negligible gain at the quartic level. AIC and BIC values also plateaued at this point, leading us to retain the cubic model as the most parsimonious choice. For the forehead, a clear improvement was observed only with the inclusion of the quartic term (adjusted R2 = 0.11; ΔAIC ≈ –270 relative to the cubic fit), indicating a mild but genuine nonlinear component. The cheeks showed consistent but substantially smaller improvements, leveling off at the cubic-quartic range (adjusted R2 ≤ 0.10, ΔAIC: 15–40). These results demonstrate that the nose contributes the dominant, largely monotonic signal; the forehead and cheeks add weaker, region-specific nonlinear information; and higher-order terms were retained only when they reduced AIC or BIC by more than ten points, thereby avoiding over-parameterization. The final best-fit curves are overlaid on the frame-level data in Figure 7 (p. 12), providing an intuitive visualization of the quantitative results. The corresponding explanatory text has been added to p. 10 of the revised manuscript.
Point 7: Uncontrolled Confounding Factors: Potential confounding variables known to affect facial-skin temperature were not adequately controlled or included as covariates in the model. These include:
- Subtle fluctuations in the reported ambient temperature (23.5–24.5 °C).
- Individual differences in baseline skin temperature and vascular tone.
- Respiration patterns and other unmonitored physiological processes (e.g., subtle muscle activity not related to emotion)..
Response: We acknowledge the point that factors such as ambient temperature fluctuations, baseline skin temperature, respiration, and subtle muscle activity were not explicitly modeled. While the room temperature was maintained within a narrow range (23.5–24.5 °C), and all participants followed the same protocol in a controlled environment, we agree that these variables may still introduce residual variability in facial thermal signals. In the revised Discussion (p. 14), we have added a note highlighting this limitation and suggesting that future work incorporate physiological covariates (e.g., baseline skin temperature, respiration rate, facial EMG) to account more accurately for potential confounds.
Point 8: Participants perform valence ratings and arousal ratings after watching the films. However, the text does not define these two indicators. Additionally, is the evaluation method for valence ratings and arousal ratings consistent for the five different emotions?
Response: We have clarified the definitions of valence and arousal on p. 2 and now explicitly state that participants reported both dimensions using consistent self-assessment procedures across all five film clips (p. 2).
Point 9: The approach of having participants manually select valence and arousal ratings is overly subjective. Can EEG (electroencephalogram) or neurotransmitter detection be used as ground truth to verify the accuracy of valence and arousal ratings?
Response: We agree that relying solely on self-reported valence and arousal is a limitation, as such ratings may contain individual biases and cannot be cross-validated against physiological ground truth. Nevertheless, self-assessment remains the standard reference in affective-science research and has been shown to correlate reliably with validated emotional film stimuli. All participants followed the same scoring procedure under identical viewing conditions, which helps reduce systematic bias across the five clips. We have now acknowledged this limitation in the revised Discussion (p. 14) and noted that future work will incorporate concurrent physiological measurements (e.g., electrodermal activity, heart rate variability, EEG) to provide objective validation and enable multimodal modeling of thermal and physiological signals.
Point 10: In 3.2, it is recommended to add a comparison curve between the real and predicted dynamic changes in subjective valence and arousal.
Response: As suggested, we have provided representative predicted-versus-actual curves for the ResNet-34 model in Figure 5.
Point 11: The manuscript describes continuous thermal recording and continuous valence/arousal rating but does not explicitly address the temporal correspondence between the two data streams. If emotional fluctuations occur during movie viewing, thermal data from that period should correspond to concurrent rating changes; no such time-series correlation analysis is mentioned.
Response: As suggested, we have described the procedure used to synchronize the film presentation with thermal imaging and rating data acquisition in the Methods section (p. 4).
We thank you for your helpful and constructive critique of our manuscript. We hope that the revised manuscript is now acceptable for publication in Sensors.
Yours faithfully,
Wataru Sato
Reviewer 2 Report
Comments and Suggestions for Authors
This paper presents a well-designed study on the application of machine learning (ML) for dynamic emotion sensing using facial thermal imaging. The research addresses significant limitations of traditional linear methods and ROI-based analyses, offering novel contributions to emotion recognition technology. The manuscript is logically structured, methodologically rigorous, and clearly written, with strong potential for real-world applications in mental health and human-computer interaction. There are some comments:
- The LOPOCV method was used in the paper, but in some places it was written as LOOCV. Please check it carefully.
- In Fig.6, authors said:” The curves for the left and right cheeks were almost identical, showing a high level of symmetry, which suggests consistent autonomic activity between both sides during emotional arousal.” However, the numerical values of the vertical axes of the two graphs differ significantly.
- There are both valence and arousal dynamic ratings in Fig.3, while there is only arousal rating in Fig.6. Why?
- From the curve of valence dynamic rating in Fig.3, it can be seen that the “angry” curve is very different from other curves. The authors can give some explanations.
Author Response
Dear Sir,
Thank you for your email dated June 26, 2025, in which you kindly advised us to revise our manuscript (sensors-3716574). We have carefully revised the manuscript based on your suggestions. The major changes made to the manuscript are indicated in red font in the tracked version. Additionally, a professional English proofreading service made some minor edits (http://www.textcheck.com/certificate/lTTyPc); these changes are not highlighted unless the content itself was modified.
Point 1: The LOPOCV method was used in the paper, but in some places it was written as LOOCV. Please check it carefully.
Response: Thank you for pointing this out. We have carefully reviewed the manuscript and corrected all instances of “LOOCV” to “LOPOCV”, to ensure consistency and accuracy throughout the text.
Point 2: In Fig.6, authors said: "The curves for the left and right cheeks were almost identical, showing a high level of symmetry, which suggests consistent autonomic activity between both sides during emotional arousal." However, the numerical values of the vertical axes of the two graphs differ significantly.
Response: As suggested, we have revised the text on p. 12 to clarify that, although the left and right cheek curves exhibit similar shapes, their attribution magnitudes differ, specifically the right cheek shows slightly higher values. This revised phrasing more accurately reflects the data presented in new Figure 7.
Point 3: There are both valence and arousal dynamic ratings in Fig.3, while there is only arousal rating in Fig.6. Why?
Response: As suggested, we have outlined our rationale for analyzing arousal ratings in the Introduction (p. 3).
Point 4: From the curve of valence dynamic rating in Fig.3, it can be seen that the “angry” curve is very different from other curves. The authors can give some explanations.
Response: As suggested, we have added a clarification on p. 7 explaining that the sharp change in valence observed for the “angry” condition reflects the structure of the film, which begins with pleasant content and later transitions to an anger-inducing scene.
We thank you for your helpful and constructive critique of our manuscript. We hope that the revised manuscript is now acceptable for publication in Sensors.
Yours faithfully,
Wataru Sato
Reviewer 3 Report
Comments and Suggestions for Authors
In my opinion, the study "Development of Machine Learning-based Facial Thermal Image Analysis for Dynamic Emotion Sensing" is relevant and has a fundamental significance to some extent. With the advancement of machine learning methods, there is an opportunity for a deeper exploration and prediction of emotional arousal based on thermal data from a person's face. The model utilizing artificial intelligence demonstrated better results than the traditional linear regression methods previously used.
The work aligns with the theme of the journal Sensors and presents novelty in the approaches to analyzing human emotions based on thermal data from a person's face. The analysis of temperature changes is conducted using neural networks, which allows for deeper connections between temperature variations and emotional arousal, as the approximation is not limited to linear functions. The authors need to justify their choice of the ResNet-34 neural network. Additionally, with a small sample size, there is a high probability of overfitting the network; it may be necessary to present results with different numbers of training epochs.
The authors provide well-reasoned references to previous works, and the cited studies are recent, indicating the relevance of the research. The conclusions align with the stated objectives at the beginning of the work. However, in the conclusions or in the discussion chapter, it should be indicated that there is potential for using other neural networks (for example, the YOLO network, which performs very well with images). It is rightly noted that a limitation of the study is the small sample size (a limited number of participants in the experiment). Increasing the number of participants can lead to different outcomes, distort the results, or improve the accuracy of predictions?
The study employs high-tech equipment from American and Japanese manufacturers, and the experimental conditions are sufficiently accurate (maintaining a constant temperature in the room, participants being of similar age groups, etc.). The research methods are also validated in other fields (the ResNet-34 network is widely used for outcome prediction). This allows us to assert the correctness of the experiment and the obtained results.
The graphs in Figure 6 present third-degree polynomial regression lines; however, I would not say that the distribution of points belonging to the Forehead region closely follows this curve. It may be necessary to use different curves for different regions of the face to represent the relationships between temperature and arousal ratings. Perhaps a second-degree polynomial would suffice, or introducing a bias could make an exponential relationship more suitable. If possible, please clarify the choice of third-degree polynomial regression lines for all regions.
There is a small inaccuracy: in Figure 5, you refer to Figure 2 (right), but there is only one image there.
Author Response
Dear Sir,
Thank you for your email dated June 26, 2025, in which you kindly advised us to revise our manuscript (sensors-3716574). We have carefully revised the manuscript based on your suggestions. The major changes made to the manuscript are indicated in red font in the tracked version. Additionally, a professional English proofreading service made some minor edits (http://www.textcheck.com/certificate/lTTyPc); these changes are not highlighted unless the content itself was modified.
Point 1: The authors need to justify their choice of the ResNet-34 neural network.
Response: As suggested, we have now described our rationale for machine learning model selection in the Introduction (p. 2). We selected the ResNet-34 model as a representative example of deep learning models.
Point 2: Additionally, with a small sample size, there is a high probability of overfitting the network; it may be necessary to present results with different numbers of training epochs.
Response: Thank you for raising the concern about potential overfitting. For every fold in the LOPOCV procedure, we recorded the full training and validation MSE trajectories and saved the checkpoint that achieved the minimum validation MSE. Supplementary Figure S1 now displays a set of representative learning curves (one from each fold), showing that the validation error plateaued within the first few epochs and did not increase thereafter, while the subsequent decline in training loss was only marginal. Since the reported ResNet-34 models were selected based on the lowest validation error in their respective folds, and no fold exhibited a divergence between training and validation curves, we are confident that the results shown in Figure 4 are not driven by overfitting and are stable across the range of epochs explored.
Figure S1.
Point 3: However, the discussion or conclusion section should include a note that alternative neural network architectures (e.g., YOLO, which is highly effective for image tasks) could be considered in future work.
Response: As suggested, we have now broadened our baseline model set by adding two conventional machine-learning regressors, Random Forest and Support Vector Regression, to the performance comparison in §â€¯3.2 and Figure 4. We have also added a sentence in the final paragraph of the Discussion (p. 14) stating that future work will investigate alternative deep learning architectures to enhance further predictive accuracy and interpretability.
Point 4: It is rightly noted that a limitation of the study is the small sample size (a limited number of participants in the experiment). Increasing the number of participants can lead to different outcomes, distort the results, or improve the accuracy of predictions?
Response: As suggested, we have now explicitly acknowledged the limited sample size as a key limitation of the study in the revised Discussion (p. 13), noting that increasing the number of participants could improve model generalizability.
Point 5: The graphs in Figure 6 present third-degree polynomial regression lines; however, I would not say that the distribution of points belonging to the Forehead region closely follows this curve. It may be necessary to use different curves for different regions of the face to represent the relationships between temperature and arousal ratings. Perhaps a second-degree polynomial would suffice, or introducing a bias could make an exponential relationship more suitable. If possible, please clarify the choice of third-degree polynomial regression lines for all regions.
Response: Thank you for raising this point. We have replaced the uniform third-degree polynomial fits with a data-driven model selection procedure: for each ROI, we now compare first- through fourth-degree polynomials using adjusted R2, RMSE, AIC, and BIC, and retain the simplest model that minimizes AIC/BIC while showing no substantial improvement with higher-order terms. As detailed in the revised text on pp. 10–11 and summarized in Table 1, this procedure resulted in a cubic fit for the nose tip and right cheek, and a quartic fit for the forehead and left cheek. Figure 7 now displays these final best-fit curves. This revision demonstrates that the forehead data indeed require a higher-order term, while other regions are adequately modeled with a cubic polynomial.
Point 6: There is a small inaccuracy: in Figure 5, you refer to Figure 2 (right), but there is only one image there.
Response: Thank you for pointing out this oversight. We have corrected the caption of Figure 5 by removing the inaccurate reference to “Figure 2 (right)” and now simply refer to “Figure 2”.
We thank you for your helpful and constructive critique of our manuscript. We hope that the revised manuscript is now acceptable for publication in Sensors.
Yours faithfully,
Wataru Sato

Reviewer 4 Report
Comments and Suggestions for Authors
The paper presented an approach to assess emotional arousal from a thermal image of a face. In the study, 20 participants watched emotional videos while thermal data was recorded. The study used several models (linear regression, random forest, SVR, ResNet) to determine the arousal value, and also used methods for interpreting models to understand which areas of the face the model pays attention to.
Strengs:
1. Innovative approach - the value of human arousal (arousal) is read using a heat map image.
2. The methodology for evaluating the results of the model is well built, various regression models and statistical tests were used.
Lacks:
1. The sample is too small - only 20 people + only young Japanese people participate in the study - raises the question of the possibility of generalizing the model to other groups
2. No multimodal validation - it would be possible to compare this method of determining by a heat map (for example, with the method of determining arousal by an image or ECG data)
the question of the correctness of retraining the ResNet model with the weights trained on ImagNet.
3. It seems to me that it would be possible to compare these models with lightweight models (MobilNet, EfficientNet).
Author Response
Dear Sir,
Thank you for your email dated August 15, 2025, in which you kindly advised us to revise our manuscript (sensors-3716574). We have carefully revised the manuscript based on your suggestions. The major changes made to the manuscript are indicated in red font in the tracked version.
Point 1: The sample is too small - only 20 people + only young Japanese people participate in the study - raises the question of the possibility of generalizing the model to other groups.
Response: We agree with this important point. As noted in the Discussion (p.13), we acknowledged that the sample size was relatively small and demographically homogeneous, which may restrict generalizability. We also stated that future studies should recruit larger and more diverse cohorts to validate the robustness of the model across different populations.
Point 2: No multimodal validation - it would be possible to compare this method of determining by a heat map (for example, with the method of determining arousal by an image or ECG data) the question of the correctness of retraining the ResNet model with the weights trained on ImagNet.
Response: We appreciate the reviewer’s comment regarding the absence of multimodal validation. As revised in the Discussion (p.14), our analysis relied exclusively on thermal facial maps, which may limit robustness. In line with the reviewer’s suggestion, future work will perform multimodal validation by directly comparing heat-map–based arousal estimates with visible-spectrum image features and physiological signals (e.g., ECG, EDA, EEG) to provide independent benchmarks beyond self-reports.
Point 3: It seems to me that it would be possible to compare these models with lightweight models.
Response: As noted, we also explored lightweight architectures such as MobileNet during our model selection process. Their performance differences were marginal, and therefore we ultimately selected the regression model with a ResNet-34 backbone for the main analyses based on its superior accuracy. We have clarified this point in the revised manuscript (p.6).
We thank you for your helpful and constructive critique of our manuscript. We hope that the revised manuscript is now acceptable for publication in Sensors.
Yours faithfully,
Wataru Sato